# Exploring and Mapping Screening Tools for Cognitive Impairment and Traumatic Brain Injury in the Homelessness Context: A Scoping Review

**DOI:** 10.3390/ijerph20043440

**Published:** 2023-02-15

**Authors:** Erin M. Fearn-Smith, Justin Newton Scanlan, Nicola Hancock

**Affiliations:** Faculty of Medicine and Health, Centre for Disability Research and Policy, The University of Sydney, Camperdown, NSW 2050, Australia

**Keywords:** homelessness, cognitive impairment, traumatic brain injury, screening tools, case management, rehabilitation, recovery, supportive housing, social determinants of health

## Abstract

Cognitive impairment is common amongst people experiencing homelessness, yet cognitive screening and the collection of history of brain injury rarely features in homelessness service delivery practice. The purpose of this research was to scope and map strategies for screening for the potential presence of cognitive impairment or brain injury amongst people experiencing homelessness and identify instruments that could be administered by homelessness service staff to facilitate referral for formal diagnosis and appropriate support. A search was conducted across five databases, followed by a hand search from relevant systematic reviews. A total of 108 publications were included for analysis. Described in the literature were 151 instruments for measuring cognitive function and 8 instruments screening for history of brain injury. Tools that were described in more than two publications, screening for the potential presence of cognitive impairment or history of brain injury, were included for analysis. Of those regularly described, only three instruments measuring cognitive function and three measuring history of brain injury (all of which focused on traumatic brain injury (TBI)) may be administered by non-specialist assessors. The Trail Making Test (TMT) and the Ohio State University Traumatic Brain Injury Identification Method (OSU TBI-ID) are both potentially viable tools for supporting the identification of a likely cognitive impairment or TBI history in the homelessness service context. Further population-specific research and implementation science research is required to maximise the potential for practice application success.

## 1. Introduction

Homelessness is a growing global health concern, with increasing international commitment to addressing the problem [1]. The risk of becoming and remaining homeless can be significantly increased by a cognitive impairment [2], and the prevalence of cognitive impairment is much higher in homeless populations [3,4,5]. The primary causes of cognitive impairment in homeless populations have been attributed to developmental disability, psychotic illness, substance use, traumatic brain injury (TBI), and other neurological conditions [6]. Details pertaining to most of these issues are often routinely collected in homelessness service delivery, including disability status, psychiatric diagnoses, substance use behaviours, and diagnosed health conditions of people accessing the service. However, despite recommendations [5,6,7], collecting history of brain injury and screening for cognitive impairment are not standard practices in homelessness service delivery [8].

High rates of cognitive impairment [5,9,10,11], brain injury [12,13,14,15,16,17,18,19,20,21,22,23,24,25,26,27], or both [7,28,29] have been found in homeless populations. Failure to identify these issues amongst people experiencing homelessness affects other social determinants at the individual level, including poor housing, health, social, vocational, and justice outcomes for people in this cohort, places undue pressure on the homelessness service delivery system, and increases demand on emergency and other public service responses.

Cognitive impairments, resulting from brain injuries or other causes, are thought to play a role in limiting housing success [30,31]. Homeless populations have been found to have substantial deficits in memory, attention, learning, cognitive processing speed, and executive function [3,32], and this has been associated with functional limitations, including poor problem solving and independent living skills, increased risk taking and interpersonal conflict, and reduced community participation and income [33,34,35,36].

A scoping review conducted by Stone et al. [2] highlights that in addition to increased rates of cognitive impairment among people who are homeless, risks associated with becoming or remaining homeless are exacerbated by socioeconomic factors. As a group who are often less engaged with employment, education, and regular planned and preventive health and welfare services [15,29,37], poorer health and housing outcomes are common [3,4]. While improvements in cognitive function have been observed upon resolution of homelessness [38,39], ongoing support needs have been found amongst those with a diagnosed cognitive impairment [40]. Securing appropriate support services to coincide with housing acquisition is, therefore, likely to improve housing sustainability for people exiting homelessness with a cognitive impairment.

In a study that looked at case manager estimations of cognitive function amongst their homeless clients, Vella [41] found that prior to screening, cognitive and functional capacity of people accessing homelessness services was overestimated by service staff, resulting in unrealistic expectations of independent living success. This may lead to inadequate referrals for specialist services and housing choices that are not sustainable without support, and in turn may be associated with repeat use of homelessness services [41]. Another study found that staff both over- and underestimate homeless clients’ cognitive capacity and that homelessness service providers care for substantial numbers of people with memory problems, cognitive decline, and dementia who are not receiving the necessary specialist services [42]. This increases pressure on the homelessness service system and reduces overall capacity to deliver services to people without complex needs. This increased demand may, therefore, contribute to significant numbers of people being unable to access timely services, as has been reported in the US [43], Canada [44], and Australia [45].

In addition to poor individual outcomes, inadequate engagement with planned health and welfare services, and inefficient utilisation of the homelessness service system, long-term homelessness is associated with increased use of emergency and other public resources. People experiencing homelessness, cognitive impairment, and TBI are more likely to have contact with police, fire service and paramedics, increased presentations to emergency departments, greater frequency of hospital admissions, high use of emergency housing and welfare services, and are more likely to be in contact with state justice and child protection [24,46,47]. A similar profile exists for people with cognitive impairment who are housed with inadequate support [48]. As such, there is a strong economic argument for disrupting the cycle of homelessness through the prompt identification of cognitive impairment and brain injury.

Formal assessment and diagnosis are required to open pathways to appropriate support for people with a cognitive impairment. When facilitating this, health professionals are infrequently readily available either to homelessness services, which are often under resourced, or to homeless cohorts more generally, who are often disengaged from services. Behaviour associated with brain injury or cognitive impairment is very often attributed to substance use, mental health, or ‘difficult’ behaviour. The introduction of efficient screening for cognitive impairment and brain injury, as a brief indicator of a potential issue, as opposed to a clinical examination, may offer an efficient prompt for formal assessment and diagnosis, facilitate appropriate service delivery, and attract more suitable funding streams. This would support long-term and repeat users of homelessness services to access service environments that have capability and capacity to understand the impacts of cognitive impairment or brain injury on functional impairments, as well as add new capacity to homelessness programs, and reduce the unplanned use of other public resources. 

Despite this opportunity for improved health outcomes and resource allocation, routine screening for cognitive impairment and brain injury does not feature in homelessness service delivery practice [11,22,49]. Research in this area has asserted the importance of identifying cognitive impairment and TBI [7,8,9,14,50,51,52,53], and provided helpful recommendations for clinicians in identifying cognitive impairment utilising abbreviated batteries more suited to a homeless client population [54,55], as well as guidelines for working with homeless people with an identified cognitive impairment or brain injury [11,51,56]. However, given that a great number of homelessness programs are operated by non-government organisations [57,58,59,60], a gap exists for people in this population who are being supported exclusively in non-clinical homelessness services by non-clinical staff. A system for identifying a likely cognitive impairment or brain injury to prompt referral for formal assessment is, therefore, required. This would maximise opportunities for this population to be appropriately diagnosed, treated, and supported in healthcare and disability service environments. 

With a view to improving referral pathways in practice, this study aimed to explore instruments described in the peer-reviewed literature that have been used to identify cognitive impairment or brain injury within the homeless population, specifically. Existing reviews have explored prevalence and instruments that assess either cognitive function or history of brain injury [2,26,27]. These studies found that prevalence of impairment is high in this population, great variation in aetiology exists, and this population experience comorbid health problems and have compromised functional outcomes. They also demonstrate that prevalence studies are overrepresented in the literature and that qualitative studies capturing the voices of this population are underrepresented.

Building on those works, the objective of this research was to scope and map what is contained in the literature and specifically identify who was being screened and where, the instruments regularly employed to screen for the potential presence of cognitive impairment and brain injury amongst people experiencing homelessness, and instruments that are potentially viable for implementation in homelessness service delivery practice. Considerations in viability for practical application included:Unrestricted in terms of qualifications of the person administering the screening tool.Economical in terms of cost and time resources required.Considerations of acceptability to assessors and those being assessed.Sensitive and specific to identifying a brain injury or cognitive impairment amongst people accessing homelessness services.

## 2. Materials and Methods

The design of this study is a scoping review, selected as an effective method for exploratory research into a complex topic. The purpose of the research is aligned with the common reasons outlined for undertaking this type of study [61]: to identify and map the nature and extent of research available in the literature regarding the identification of potential cognitive impairments or brain injury amongst people who are experiencing homelessness. The applied methodology reflects the framework published by Arksey and O’Malley [61], supported by JBI guidance [62], and reported following the Preferred Reporting Items for Systematic Reviews and Meta-Analyses (PRISMA) guidelines [63].

### 2.1. Identification of the Research Questions

This scoping review was designed with the intention of answering the following research questions:What tools or strategies have been employed in the literature for identifying cognitive impairment or brain injury amongst people who are experiencing homelessness?Who are the populations being assessed in the literature, and under what circumstances?Are there tools that can be administered in non-clinical homelessness service environments, by non-specialist staff, that are sensitive to identifying a likely cognitive impairment or brain injury?

### 2.2. Identification of Relevant Literature

The purpose of a scoping study is to engage in a thorough review of the available literature. Records were generated through a systematic database search and hand searching through relevant systematic reviews that were generated through the database search. 

#### 2.2.1. Search Terms

As a scoping review, this study employed the PCC mnemonic (Population; Concept; Context) described by the Joanna Briggs Institute (JBI) [64,65] for developing terms to be applied in search strategy design and record screening for scoping reviews. This study considered all types of peer reviewed research that focused on the following: Population: people experiencing or at risk of homelessness.Concept: identifying cognitive impairment or brain injury.Context: assessment, screening, or measurement.

#### 2.2.2. Database Search

A preliminary database search using 20 search terms was conducted to generate relevant key words to develop a search strategy, resulting in a total of 57 terms applied to the search. In Week 3 of January 2021, database searches were conducted in CINAHL, Embase, Medline, PsycInfo, PsycTests, Scopus, and Web of Science. Medical Subject Heading (MeSH) terms were applied; however, of the mapped terms related to the concept, specific diagnoses were not included, and mapped terms related to the context did not include specific tools. Terms related to the population were combined with ‘OR’, as were terms related to the concept and the context. The collated results of these three searches were combined with ‘AND’. The search strategy is available in Appendix B Table A1 ‘Search strategy’.

Database search results were exported to Endnote, and records were exported to Covidence for screening against inclusion criteria. Relevant studies were mapped in Microsoft Excel.

#### 2.2.3. Hand Search

Although systematic reviews were not included in the study, further relevant publications were sourced by hand searching the systematic reviews that were generated from the database search.

### 2.3. Selection of Studies

Literature was screened by two reviewers using online Covidence software [66], and sources were assessed, by title and abstract, and then full text, against the inclusion criteria (see Table 1. Inclusion and exclusion criteria). Papers for inclusion described screening for any potential impairment in cognitive functioning or history of brain injury amongst people who were homeless or at risk of returning to homelessness. As this study seeks to identify possible tools for pilot implementation in practice settings, studies describing only paediatric participants were excluded. This decision was made both because instruments screening for paediatric cognitive impairment often cannot be applied with adult populations (and vice versa), and because unaccompanied children under 16 years are usually intended to be supported through state authorities rather than homelessness services. Acknowledging that family interventions are common in homelessness service delivery, and unaccompanied minors are often neglected through state-based out-of-home care structures, investigation in this area will be recommended for future studies. 

Covidence captured any conflicts between reviewers regarding inclusion or exclusion (such as when studies included both homeless and housed populations or participants both over and under the age of 16), and consensus was reached through discussion and mutual agreement to include all studies that specifically reported results reflecting the population of interest.

#### 2.3.1. Title and Abstract Screening

Items were first assessed by title and abstract, excluding irrelevant publications. Titles with no available abstract were individually searched using the University of Sydney Cross-Search tool. Items that were not able to be located locally were ordered through inter-library loans.

#### 2.3.2. Full Text Screening

The remaining items were screened using the full text and assessed against the inclusion criteria by both reviewers. Relevant literature reviews that were generated through the database search were excluded from the study but included for hand searching potential additional papers.

### 2.4. Data Mapping

Data were mapped into the following categories: study details, sampling, instruments applied, prevalence, administered by, administration time, and acceptability to consumers and assessors. A complete data map is available as a Appendix A (Appendix A: Data Map).

### 2.5. Collation, Summary, and Report of Results

A PRISMA [63] flowchart was employed for presentation of the selection and exclusion of studies. A summary of findings from data mapping is presented in tables and discussed in relation to the research questions.

## 3. Results

### 3.1. Selection of Studies

The PRISMA diagram in Figure 1 demonstrates the data identification and screening process that informed the selection of studies. A total of 108 publications were included in the final review (available as Table A2 in Appendix C ‘Literature chart’).

### 3.2. Extracting and Charting

The literature was mapped against fields recommended by the JBI (2017): author(s), year of publication, source origin/country of origin, study population and sample size, study setting, instruments applied, administration, screening outcomes, duration of assessment, how outcomes are measured, and key findings that relate to the review questions.

Included papers ranged in publication date between 1990 and 2021, and included 12 dissertation publications and 96 peer-reviewed articles. Articles spanned 66 journals, with a median of 1 publication per journal: 48 journals published 1 study, 13 journals published 2 studies, and only 5 journals published 3 or more studies in this area of research. 

#### 3.2.1. Populations and Contexts

Literature summary features are shown in Table 2. Of the focus populations across the studies, though there was great variability in participant age, setting, and co-morbid diagnoses, the majority of people represented in the literature were men in Western countries. Almost three-quarters of the papers reported studies conducted in North America (USA, n = 56; Canada, n = 23), and most of the remainder reflected populations in Western, English-speaking countries. Men were most heavily represented in participant populations, with 99 of the 108 papers comprising more than 50% male participants.

All studies included screening people experiencing homelessness; however, not all studies were conducted exclusively with this population, or in settings exclusive to this population. Ninety-four studies were conducted in settings exclusive to people who are homeless: crisis accommodation or shelter environments (n = 42), generalist homelessness or outreach support settings (n = 31), post-crisis housing support (n = 11), and health settings for people who are homeless (n = 10). Fourteen studies were conducted in other health or community settings that service both people who are and who are not homeless, including general health or hospital settings (n = 7), mental health or substance use treatment programs (n = 5), and diverse community or social welfare settings (n = 2). The focus populations also varied across the literature, with just over half of the study populations being homeless adults (40% mixed, 9% men only, 4% women only), and the remaining studies focusing on populations with additional criteria for inclusion, such as age, cultural group, mental health status, substance use, veterans, or co-/multimorbidity (see Table 3). The majority of studies focused on measuring cognitive function (n = 87), though 31 of those papers also reported the presence of brain injury amongst participants. The remaining 21 studies focused exclusively on screening for brain injury. 

A number of issues relevant to this group were regularly reported across the literature. Participant data were often inclusive of issues pertaining to substance use (n = 85), schizophrenia or other psychotic illness (n = 50), diagnosed mental illness (specified, other than psychotic illness) or undefined mental illness (n = 82), length of homelessness considered (n = 56), frailty or health conditions (n = 28), veteran (n = 19), developmental disability (n = 15), adverse childhood experiences (n = 11), history of incarceration (n = 10), and domestic and family violence (n = 4).

#### 3.2.2. Screening Instruments

Across the 108 publications, a total of 158 separate instruments were described, including individual components of a single assessment tool and various versions of the same instrument component (see Appendix D, Table A3 ‘Complete list of screening instruments’). Of the 158 instruments, 151 were cognitive screens and 7 were TBI screens.

After grouping together identical instruments (including unchanged components of updated assessment tools), reviewers found 17 instruments measuring cognitive function and 3 screens for brain injury (all of which focused only on screening for TBI) that appeared in more than two publications, and that were used at least twice to explicitly screen for the presence or absence of cognitive impairment or TBI, as opposed to reporting a collated result to provide population-based performance data (see Table 3). 

Only 65% of those studies included details of who was administering the assessments, and of those, sometimes specialist training was implicit in the description (e.g., ‘psychiatrist’) and sometimes it was unclear (e.g., ‘research assistant’ or ‘case manager’). Further investigation was undertaken to determine whether specialist qualifications were required to administer the instruments, by first searching Pearson Assessments [67] or PAR Inc. [68] for qualification grade, and if unavailable on these platforms, through developer websites. This process determined that 14 of the 17 cognitive screening tools require specialist qualifications to administer them (psychologist, occupational therapist, or ‘health professional’), whilst the remaining 3, as well as the 3 TBI screens, are not restricted. Each of the unrestricted tools are also free to access. 

#### 3.2.3. Instruments Unrestricted by Qualification

To consider potential viability for practice implementation, the features of unrestricted screening approaches were mapped. The most commonly employed instrument for assessing cognitive function was the Mini Mental State Examination (MMSE) [69], followed by the Trail Making Test (TMT) [70] Parts A (TMT-A) and B (TMT-B). The most common approach to determining the presence of a TBI was a single or series of questions regarding history of head trauma, followed by The Ohio State University TBI Identification Method (OSU TBI-ID) [71] and the TBI-4 [72].

There was great variability between studies regarding the measured prevalence of cognitive impairment or TBI in the participant population. Time to administer was usually reported as a collated total time to administer a battery rather than time for the specific component. Details about sensitivity, specificity, and acceptability to consumers and assessors were rarely reported (see Table 4). 

#### 3.2.4. Acceptability of Instrument to Assessors and Consumers

Consideration was given to the acceptability of tools in a very small number of papers. In total, 4 of the 108 included publications considered the acceptability of the assessment process to both assessors and consumers, 1 further study considered acceptability to consumers only, and another considered acceptability to assessors only. 

Of the screening tools that appeared regularly in the literature and can be employed by non-specialist assessors, the acceptability to both consumers and assessors was considered for the MMSE and Trail-Making test Parts A and B in one single study [54], and the acceptability for assessors was considered for the OSU TBI-ID in one study [22]. Considerations included: time required; incorporation into routine appointments; both assessor and client preferences for time of day, with consideration of substance use or medication effects; demands of the assessment influencing willingness of consumer engagement; practicality in administration and interpretation; and the cost-effectiveness of the tool.

#### 3.2.5. Sensitivity and Specificity

Sensitivity and specificity were rarely described; consideration of sensitivity and specificity of the unrestricted instruments was described, or implied, in four studies. Under circumstances where they were described, it was unclear whether this reflected validated administration with homeless populations.

##### Cognitive Screens

Mini Mental State Examination (MMSE). Gash [73] described strong sensitivity and specificity of the MMSE in identifying moderate or severe cases of dementia only. Whilst Gonzalez et al. [54] did not explicitly describe sensitivity and specificity, this study did compare MMSE results with results from a neuropsychological test battery and found that although 80% of the population were found to have a cognitive impairment using the test battery, the MMSE measured prevalence of only 35%. No studies were located that specifically reported on studies evaluating specificity of the MMSE with identifying cognitive impairments within homeless populations.

Trail Making Test—Part A (TMT-A). Sensitivity and specificity of TMT-A was not described in the included literature.

Trail Making Test—Part B (TMT-B). Though sensitivity and specificity of TMT-B was not explicitly described in the included literature, Gonzalez [54] found a close correlation between results of TMT-B and the results of an abbreviated neuropsychological test battery.

Similar to the MMSE, no studies were located that explored the specificity of TMT-A or TMT-B within homeless populations.

##### TBI Screens

OSU TBI-ID. Lafferty [17] described being unable to locate information in the literature pertaining to the sensitivity and specificity of this tool; however, Russell [20] described the OSU TBI-ID as the ‘gold standard’ in identifying TBI.

TBI-4. This is an abbreviated, adapted version of the OSU TBI-ID implemented as a preference to single-question approaches to determining TBI history [72]. Russell et al. [20] found this tool to underestimate the prevalence of TBI, compared to the OSU TBI-ID.

Single or series of questions, e.g., “Have you ever had a blow to the head?”. As this was not consistently reported and does not describe a specific measure, no assessment of sensitivity or specificity applies to this method.

## 4. Discussion

Routine screening for brain injury and cognitive impairment is likely to improve referrals for earlier diagnosis and access to health and disability services, and the results of this study have important implications for practice in homelessness service delivery. Unlike details pertaining to mental health, substance use, and other known diagnoses, screens for cognitive dysfunction or history of brain injury in homeless populations are not routinely taken. Under circumstances where screening for brain injury or cognitive impairment is undertaken, measures are mostly collected for the purposes of research, such as to estimate prevalence of injury or impairment within this population. Most instruments employed in the literature require specialist training, and those that do not are limited in explicit descriptions of consideration for the service user or assessor experience, sensitivity and specificity for the identification of brain injury or cognitive impairment, and of whether the instrument has been validated with homeless populations.

With such limited research in this area, there was little consistency in focus or approach across the included literature. Substantial variability existed regarding participant gender, age, presence of co-morbidity, or type of service engagement. Some studies included only participants with a known brain injury or cognitive impairment, and some studies excluded those with existing diagnoses, particularly dementia or developmental disability. Assessment purposes varied but were rarely in the context of routine screening in service delivery, and when screening was routine, it was in a clinical context. Administrator profiles were inconsistently reported. When the assessor was reported, this was almost invariably a trained researcher or clinician. Assessment settings were also diverse. Homelessness-specific services included street outreach, crisis shelters, homeless health services, veterans homeless services, and post-homeless supportive housing programs, whilst non-homelessness-specific settings included health clinics, hospitals, community-based mental health services, detox and rehabilitation programs, community centres, and veterans services. This leaves little room for meaningful comparisons between the cohorts, the assessors, and the settings for screening people experiencing or at risk of homelessness for cognitive impairment or brain injury.

Of the potentially viable screens, whether sensitive or specific to identifying cognitive impairment or brain injury, none were validated in homeless populations [2,3,6,26,27,71,74,75]. Probably due to its unrestricted access and common administration in this area of research, the most frequently employed assessment of cognitive function in homeless populations was the MMSE. Although this tool is a reliable and validated [76] measure for the presence of dementia, limited effectiveness for identifying other cognitive deficits in homeless populations compared to other measures has been observed [54,77]. Of the remaining cognitive screens, parts A and B of the Trail Making Test (TMT) are also sensitive to identifying the presence of dementia [78] and other cognitive impairment [79], with greater cognitive demand evidenced with Part B [80,81]. Though this instrument has been described as a reliable indicator of cognitive impairment [79] and is evidenced to have results that are closely correlated with more comprehensive neurocognitive testing in people who are homeless [54], findings in other studies [32] have been inconsistent. However, TMT is brief, free to access, and has increased sensitivity with the inclusion of validated age- and education-adjusted cut-off scores [78,82]. It is also available in both pen-and-paper and digital formats, with possible additional cognitive processes that can be measured with digital formats of the test [83].

The most common screen for a brain injury was a single or series of questions regarding history of head trauma; however, this does not reflect a uniform approach to assessment and limits positive screens to those who are already aware that they have a brain injury and only focuses on TBI rather than the broader range of potential brain injury. Of the remaining TBI screens, the TBI-4 is a validated predictor of hospital admission in veteran populations [84], though less effective than the OSU TBI-ID method in identifying history of TBI amongst people who are homeless [20]. The OSU TBI-ID is a brief, valid, and reliable screen for history of TBI [74,85] that was developed based on TBI surveillance recommendations made by the Center for Disease Control and Prevention (CDC) [71]. Reporting errors are limited through the inclusion of head and neck injuries, length (if relevant) of loss of consciousness, number of injuries, age of first injury, and experience of dizziness and memory loss. Russell et al. [20] described the OSU TBI-ID as the ‘gold standard’ of TBI identification, and the instrument has been validated with military personnel, veterans, individuals with substance use, and prisoners or those with a history of incarceration [74,75,86]. The papers employing the OSU TBI-ID in this study did not specify whether they used the short, clinical, or research version of the instrument; however, the developers [71] describe the short version as being administered in 5 min. The OSU TBI-ID is also free to access.

Whilst engaging non-specialist staff in the screening process offers the promise of new referral prompts and better health and housing outcomes, potential consequences for the administration of cognitive screening tools by non-trained clinicians must also be considered. Application of screening in service settings carries the risk of being introduced as a barrier to service access, with identified scores required for inclusion or exclusion for service delivery. This could result in people with perceived impairment being excluded from services, remaining homeless until they decompensate and need institutionalised care, or experiencing early mortality. Recommendations for cognitive screening protocols in service delivery environments, therefore, include clear differentiation between screening and assessment, to ensure that non-specialist staff are clear that they are not undertaking assessment, but rather seeking to provide screening results in referrals for formal assessment. A matrix or decision tree to explicitly facilitate referral to appropriate services in the local context may also support meaningful and consistent referral outcomes.

A few limitations of this review need acknowledgement. Because of the limited body of literature in this area, when selecting papers, no discrimination was made between types of homeless populations, including definitions of homelessness and legislative or policy contexts. There were also several publications that reported data from the same studies, potentially over-representing the application of described instruments and populations receiving services. Finally, despite the diversity of search terms included, given the inconsistency of language and keywords in the literature, some relevant studies may not have been included.

## 5. Conclusions

Of the unrestricted instruments regularly reported in the homelessness literature, none offer an ideal screening tool for non-clinical homelessness service contexts to use to reliably identify a cognitive impairment within this population. However, the viability and value of implementing TMT in service delivery practice (either both parts, or Part-B only), as a fairly robust and time-effective screen for cognitive impairment with homeless populations, should be further explored. Similarly, the OSU TBI-ID offers a promising instrument for the identification of TBI, despite not capturing other forms of brain injury. The estimated ten minutes for completion of these two assessments may support routine implementation viability within homelessness service sites.

Further investigation is required in the pursuit for successful routine screening for cognitive impairment and TBI in homeless populations. Consultation with people who either work in or access homelessness services regarding facilitators and barriers to routine practice implementation is recommended in the development of a screening protocol that is acceptable to users. Potentially viable instruments are recommended to be validated with this population as well as further research exploring appropriate screening instruments for history of brain injuries other than TBI. As the application of brief screening instruments alone may not capture all those benefiting from more comprehensive assessment and support, the identification of other common factors amongst this cohort may be helpful in enhancing screening protocols and providing further indicators of potential impairment and referral needs in this group. The evaluation of application in service delivery within specific contexts will also determine the impact of screening in different services (e.g., shelter versus health centres) and different geographical locations. Finally, undertaking similar research to identify cognitive impairment and brain injury in paediatric populations experiencing homelessness may facilitate early intervention and prevent long-term experiences of homelessness.

## Figures and Tables

**Figure 1 ijerph-20-03440-f001:**
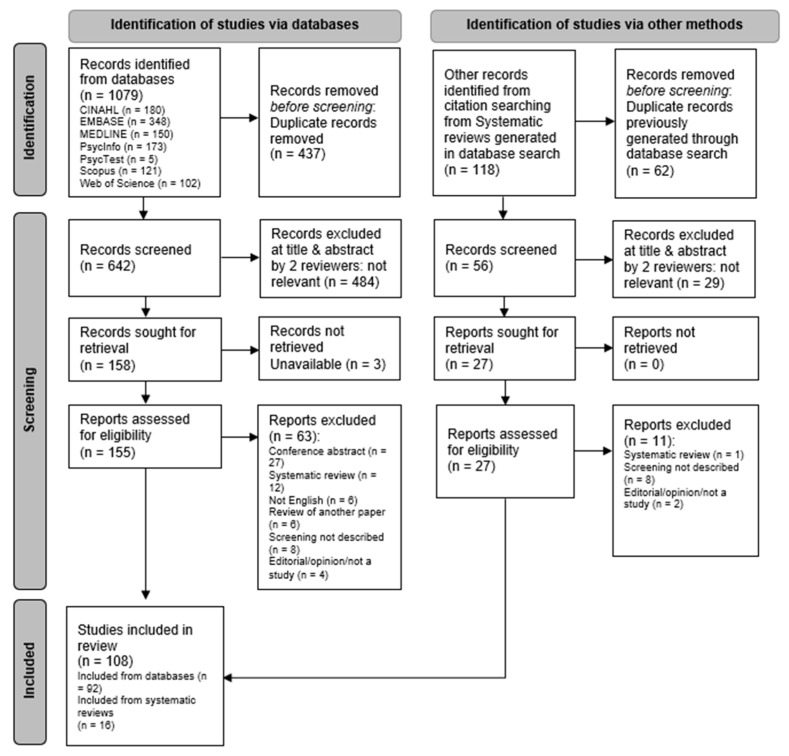
PRISMA flow diagram [63].

**Table 1 ijerph-20-03440-t001:** Inclusion and exclusion criteria.

Inclusion Criteria	Exclusion Criteria
Screening for identification of any potential impairment in cognitive functioning or brain injury amongst people aged 16+ years who are homeless (or exiting homelessness, or at risk of returning to homelessness) described. Peer-reviewed studies (incl. dissertations) Primary research Written in English	Describe only populations aged <16 years Conference abstracts or poster presentations Editorials, opinions, reports, or publications that are not peer-reviewed primary research studies Unavailable in full text Systematic reviews: *reviewers to hand search for potentially suitable papers*

**Table 2 ijerph-20-03440-t002:** Literature Summary Features.

Summary Features	n (%)
**Country**	
USA	56 (52%)
Canada	23 (21%)
UK	12 (11%)
Australia	4 (4%)
Japan	4 (4%)
Other ^^^	9 (8%)
**Screen**	
Cognitive function only	56 (52%)
Presence of brain injury only	21 (19%)
Both described	31 (29%)
**Instruments**	
Single measure	38 (35%)
Battery	70 (65%)
**Study Populations**	
Adults (general n = 43; Men only n = 10; Women only n = 4)	57 (53%)
Mental Health (MH)/Substance Use Disorder (SUD)	20 (19%)
Older People (50+)	11 (10%)
Veterans (General n = 5; Veterans with MH/SUD n = 3)	8 (7%)
Young People (16–24)	6 (6%)
Co- or Multi-morbidity	4 (4%)
CALD/BME	1 (1%)
Indigenous	1 (1%)
**Study Participants**	
50%+ male participants	99 (92%)
50%+ female participants	7 (6%)
Unknown/not specified	2 (2%)
**Study Settings**	
Homeless-specific settings	94 (87%)
Other community settings	14 (13%)

Notes. n = number; ^^^ incl. Brazil, Ecuador, Germany, Hong Kong, Israel, Mozambique, Spain, and Netherlands.

**Table 3 ijerph-20-03440-t003:** Instruments regularly screening for the presence of brain injury or cognitive impairment.

Instrument	Administration Restriction	Purchase/Administration Cost
**Cognitive screening instruments**		
Allen Cognitive Level Screen-2000/ACLS-5	Specialist ^a^	Yes
Connors’ Continuous Performance Test (CPT-II)	Specialist	Yes
Delis–Kaplan Executive Function System (D-KEFS) Tower	Specialist	Yes
Delis–Kaplan Executive Function System (D-KEFS) Trail Making Number-Letter Switching	Specialist	Yes
Delis–Kaplan Executive Function System (D-KEFS) Verbal Fluency-Category Switching	Specialist	Yes
Hopkins Verbal Learning Test-R (HVLT-R)	Specialist	Yes
Montreal Cognitive Assessment (MoCA)	Specialist ^a^	Yes
Rey Complex Figure Test (RCFT) copy trial	Specialist	Yes
Rey Complex Figure Test (RCFT) delayed recall	Specialist	Yes
Wechsler Abbreviated Scale of Intelligence (WASI)	Specialist	Yes
WASI Vocabulary subtest	Specialist	Yes
WAIS-R/WAISIII/WAIS-IV Digit Span	Specialist	Yes
WAIS-R Digit Symbol/WAIS-III Digit-Symbol coding/WAIS-IV Coding	Specialist	Yes
Wechsler Test of Adult Reading (WTAR)	Specialist	Yes
Mini Mental State Examination (MMSE)	None	No
Trail Making Test-Part A	None	No
Trail Making Test-Part B	None	No
**TBI Screening instruments**		
OSU TBI-ID	None	No
TBI-4	None	No
Single or series of questions, e.g., “Have you ever had a blow to the head?”	None	No

Note. ^a^ No restriction; however, developer recommends restriction to health professionals.

**Table 4 ijerph-20-03440-t004:** Key features of unrestricted screens.

	Cognition Screens			TBI Screens		
	Mini Mental State Examination (MMSE)	Trail Making Test-Part A (TMT-A)	Trail Making Test-Part B (TMT-B)	Ohio State University Traumatic Brain Injury Identification Method (OSU TBI-ID)	TBI-4	Single/Series of Questions, e.g., “Have You Ever Had a Blow to the Head?”
**Papers Describing the Measure**						
n	20	8	11	10	3	14
**Positive Screen**						
median	24%	44%	38%	90%	73%	58%
mean	24%	50%	48%	82%	73%	55%
range	0–78%	27–80%	25–80%	53–91%	59–87%	9–84%
*(collated result) ^1^*	*(5%)*	*(63%)*	*(45%)*	*(20%)*	*(33%)*	*(0%)*
**Time to Administer**						
range	? ^1^	? ^1^	? ^1^	? ^1^	? ^1^	? ^1^
*(collated result) ^1^*	*(90%)*	*(88%)*	*(82%)*	*(100%)*	*(100%)*	*(86%)*
**Client Acceptability Considered**						
times described	1	1	1	0	0	0
**Assessor Acceptability Considered**						
times described	1	1	1	1	0	0
**Sensitivity and Specificity of Instrument Described**						
times described	1 ^2^	0	1 ^2^	1 ^3^	1 ^3^	NA

Notes. n = number; ^1^ studies presenting a result that is a global score or reflects total time for a test battery; ^2^ described explicitly in one study; however, sensitivity also described in a second study, comparing MMSE results and TMT-B to neuropsychological battery results; ^3^ one study explicitly mentions sensitivity and specificity, in the context of being unable to find this information.

## Data Availability

The data presented in this study are not publicly available.

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
