# Peer review of "Exploring and Mapping Screening Tools for Cognitive Impairment and Traumatic Brain Injury in the Homelessness Context: A Scoping Review"

_ijerph, 2023, doi:10.3390/ijerph20043440_

Round 1

Reviewer 1 Report

Thank you for the opportunity to review this manuscript submitted for publication. Generally, I think it is a relevant and important topic to the field of homeless services, especially with a greater focus on cognition and brain injury. I appreciate the focus of identifying tools and who is able to use the tools, given the variety of providers PEH will encounter when accessing services. The manuscript follows JBI methodology for a scoping review which is clearly described.

However, I do have a few comments and recommendations:

·       For the review process, what was the process for achieving consensus if 2 reviewers didn’t agree?

·       At the introduction, I would like the authors to speak more to the Stone et al. scoping review that looked at cognition in people experiencing homelessness, as that scoping review also included what cognition and TBI screening measures were used in those studies. I think it’s important to share how this article builds on this work aside from being completed 4 years later and thus including what has been published since that time. I appreciate the additional focus of this article on mapping existing tools, cost and who can implement tools as this has really practical implications. However, I do think it’s important to reference initial work aside from placing the context of this article regarding the prevalence of cognition in the population.

·       I also noted the BISQ was included in the Appendix B in more than 2 studies, but this was not included in the table within the paper regarding the tools. Although not free, it has been used and should be included in the initial table based on the author’s criteria.

·       Overall, I wish the discussion had done more to analyze and draw conclusions from the findings, or delve more into some of the findings. There is a bit of tendency to restate the results but there is opportunity to explore more implications for practice.  

o   For example, did any of the studies that asked questions about head injury occur after the publication of the OSU TBI ID? Has the use of the OSU has replaced the open questions (as noted it is now the recommended gold standard), and I think this would be an important shift to highlight (if it exists or not).

o   Are there potential consequences for administration of cognitive screening tools by non-trained clinicians, and how should these findings be used? Within practice, I have unfortunately seen results of cognitive screenings (without interpretation/application) be incorrectly used as a barrier to a person accessing services (e.g., the person was declined from permanent supportive housing). This is a very serious consideration, as many communities do not have a tiered approach to housing and supports, and perceived impairment excludes individuals from services but keeps them homeless unless they decompensate and need institutionalized care, or experience early mortality (this is unfortunately not a unique or singular experience). I think it’s important to explore the role of screening for cognition and perhaps make recommendations that this creates referrals to services and not as a barrier. This may seem implied but unfortunately is not. (Or the other argument could be that people experiencing homelessness have access to specialized services that can adequately assess for cognitive needs).  

o   This should also share more about needed research and next steps for the field. Do we need to evaluate tools within specific contexts to see what is most relevant based on the role of different services (e.g. shelter versus health centers) and appropriate based on staffing knowledge? Do we need to evaluate and validate tools with this population? How does use of these tools coincide with what we know about education and literacy levels which can skew results on cognitive screens?

o   It is also noted that there is a limitation that the OSU only assesses TBI and does not include ABI – I am not sure there is enough presented at the beginning of the article to share why assessing for ABI and TBI is important (vs. just TBI). ABI is a very broad definition with a multitude of causes, which should be defined early on – why does this also need to be assessed?

o   It may also be helpful for the authors to add into the research findings of the main tools discussed in this paper regarding psychometric properties and testing on the population, versus relying on the report of the authors that used it. It is noted later in the discussion who the OSU has been tested on, but not the MMSE and TMT. This would be a helpful addition.

·       I recommend double checking and copy-editing Appendix B (the larger table of studies and findings). I noted a few errors (typos, abbreviations) and in data (e.g. Synovec 2020 states the MMSE was used, but this is incorrect). The WHO-DAS 2.0 was also included in this table, but does not explicitly measure cognition.

Although not required, it would also be helpful to include the reference for the original authors of the tools included, so that the reader could easily find and review the tools.

Author Response

Dear Reviewers,

Thank you very much for taking the time to review this paper and providing your feedback and recommendations. Please see responses to each reviewer's comments below, and the amended paper with tracked changes attached. 

REVIEWER 1

COMMENT: Thank you for the opportunity to review this manuscript submitted for publication. Generally, I think it is a relevant and important topic to the field of homeless services, especially with a greater focus on cognition and brain injury. I appreciate the focus of identifying tools and who is able to use the tools, given the variety of providers PEH will encounter when accessing services. The manuscript follows JBI methodology for a scoping review which is clearly described.

However, I do have a few comments and recommendations:

  • For the review process, what was the process for achieving consensus if 2 reviewers didn’t agree?

RESPONSE: Methods amended to include: "Covidence captured any conflicts between reviewers regarding inclusion or exclusion (such as when studies included both homeless and housed populations, or participants both over and under the age of 16), and consensus was reached through discussion and mutual agreement to include all studies that specifically reported results reflecting the population of interest."

COMMENT: ·       At the introduction, I would like the authors to speak more to the Stone et al. scoping review that looked at cognition in people experiencing homelessness, as that scoping review also included what cognition and TBI screening measures were used in those studies. I think it’s important to share how this article builds on this work aside from being completed 4 years later and thus including what has been published since that time. I appreciate the additional focus of this article on mapping existing tools, cost and who can implement tools as this has really practical implications. However, I do think it’s important to reference initial work aside from placing the context of this article regarding the prevalence of cognition in the population.

RESPONSE: Additional context specifically referencing the Stone study has been added to the fourth paragraph of the introduction to frame the socioeconomic conditions relevant to this population. Further reference to this study, along with another recent review by Stubbs et al (2020) and one by Topolovec-Vranic (2012) made in the final paragraph of the introduction, describing the focus and findings of these studies and how this review is different by focusing specifically on factors related to practical application.

COMMENT: ·       I also noted the BISQ was included in the Appendix B in more than 2 studies, but this was not included in the table within the paper regarding the tools. Although not free, it has been used and should be included in the initial table based on the author’s criteria.

RESPONSE: Thank you so much for picking up this typo! The BISQ was actually only described in two studies, and the BISI was also described in two studies, but the BISI was recorded in both appendices B and C as the Brain Injury Screening Questionnaire (BISI), rather than the Brain Injury Screening Index (BISI).

COMMENT: ·       Overall, I wish the discussion had done more to analyze and draw conclusions from the findings, or delve more into some of the findings. There is a bit of tendency to restate the results but there is opportunity to explore more implications for practice.  

RESPONSE: Thank you. Please see specific notes re amendments against comments below.

  • COMMENT: For example, did any of the studies that asked questions about head injury occur after the publication of the OSU TBI ID? Has the use of the OSU has replaced the open questions (as noted it is now the recommended gold standard), and I think this would be an important shift to highlight (if it exists or not).

RESPONSE: That is a good consideration! Upon review, however, the OSU TBI-ID has been referenced in studies spanning 2013-2020 and use of open questions has featured in studies spanning 1996-2020, with approximately equal use before and after 2013.

  • COMMENT: Are there potential consequences for administration of cognitive screening tools by non-trained clinicians, and how should these findings be used? Within practice, I have unfortunately seen results of cognitive screenings (without interpretation/application) be incorrectly used as a barrier to a person accessing services (e.g., the person was declined from permanent supportive housing). This is a very serious consideration, as many communities do not have a tiered approach to housing and supports, and perceived impairment excludes individuals from services but keeps them homeless unless they decompensate and need institutionalized care, or experience early mortality (this is unfortunately not a unique or singular experience). I think it’s important to explore the role of screening for cognition and perhaps make recommendations that this creates referrals to services and not as a barrier. This may seem implied but unfortunately is not. (Or the other argument could be that people experiencing homelessness have access to specialized services that can adequately assess for cognitive needs).  
  • RESPONSE: Excellent point to include, thanks! This point has now been described in the second-last paragraph of the discussion, with recommendations to include a matrix or decision tree to facilitate appropriate referrals in the local context.

  • COMMENT: This should also share more about needed research and next steps for the field. Do we need to evaluate tools within specific contexts to see what is most relevant based on the role of different services (e.g. shelter versus health centers) and appropriate based on staffing knowledge? Do we need to evaluate and validate tools with this population? How does use of these tools coincide with what we know about education and literacy levels which can skew results on cognitive screens?
  • RESPONSE: Excellent points! Thank you! Included in conclusion

  • COMMENT: It is also noted that there is a limitation that the OSU only assesses TBI and does not include ABI – I am not sure there is enough presented at the beginning of the article to share why assessing for ABI and TBI is important (vs. just TBI). ABI is a very broad definition with a multitude of causes, which should be defined early on – why does this also need to be assessed?
  • RESPONSE: This was actually a limitation identified by the developers of the tool. I have now amended language throughout the paper to reflect brain injury as our interest, as that is actually what we were looking for in terms of screening, it’s just that screening tools generated from the review were either screens for cognitive function or for TBI history exclusively (other than one study which identified a likely ARBI through combined results of a cognitive screen and a substance use screen).

  • COMMENT: It may also be helpful for the authors to add into the research findings of the main tools discussed in this paper regarding psychometric properties and testing on the population, versus relying on the report of the authors that used it. It is noted later in the discussion who the OSU has been tested on, but not the MMSE and TMT. This would be a helpful addition.
  • RESPONSE: Amendments made to describe where tools have been validated, however none have been validated with homeless populations except for MMSE, which has only been validated to detect dementia, and not mild or other forms of cognitive impairment.

COMMENT: ·       I recommend double checking and copy-editing Appendix B (the larger table of studies and findings). I noted a few errors (typos, abbreviations) and in data (e.g. Synovec 2020 states the MMSE was used, but this is incorrect). The WHO-DAS 2.0 was also included in this table, but does not explicitly measure cognition.

RESPONSE: You're right! Thank you! This was an error has been corrected and the rest of the table further reviewed.  Re the WHO-DAS 2.0, it was included because although it does not explicitly measure cognition, it does cover cognition (communication and understanding) as a domain of functioning and therefore offer indication of potential functional limitation to prompt for formal referral. This broad inclusion has been made clearer in the methods. Thank you!

COMMENT: Although not required, it would also be helpful to include the reference for the original authors of the tools included, so that the reader could easily find and review the tools.

RESPONSE: Each tool referenced with developer details (except TBI-4, which rather has reference to the paper that describes the practice application of the tool, as an abbreviated OSU TBI-ID) in 3.2.3.

REVIEWER 2

This very interesting review focused on identifying tools and strategies used to identify brain injury and cognitive impairment. There was an emphasis on understanding what tools exist that can be administered by non-clinician/specialist service providers in homeless service settings to expedite referrals to specialty services. This review identifies some important gaps in our current knowledge regarding screening for brain injury or cognitive impairments in homelessness service settings, particularly regarding the validity and acceptability of screening tools in these settings and provides a foundation upon which further research can be built. A scoping review was an appropriate choice to examine the identified research objectives, and the methodology was guided by an established framework (Arksey and O’Malley) for conducting scoping reviews and this report also adhered to PRISMA reporting guidelines. The review adheres closely to these guidelines for robust reviews.

Feedback for manuscript overall. My main feedback/concerns overall are related to clarification of terms and concepts used throughout the report. Specifically:

COMMENT: Cognitive impairment has a wide scope as a concept and it is not clear in the introduction as a reader why CI is being examined along with TBI specifically, as opposed to cognitive impairment along with acquired brain injury more broadly. There is also reference to acquired brain injury or brain injury more generally in other sections of the manuscript, including the inclusion criteria. As homeless persons are at high risk for non-traumatic ABI in addition to TBI (e.g. hypoxic brain injury related to substance overdose) some clarity about the focus on TBI or whether all acquired brain injuries fell within the scope of this review is needed.

RESPONSE: Thank you, this was tricky, as the review actually included search terms for ABI more broadly, but the literature exclusively reflected tools to discover TBI history, or measures to identify cognitive impairment (except for a single study utilised a combination of substance use information and cognitive screening to determine alcohol-related brain injury, but still the tool was a cognitive screen. Language now changed to reflect brain injury rather than TBI throughout the paper.

COMMENT: The terms screening, assessment, and measurement are used throughout the manuscript and I wonder if some clarity about the concept of screening versus assessment might be helpful. Screening would appear to be the focus of this review (ie. a brief test for early identification of an issue, often part of routine care, can be administered by non-clinician staff with appropriate training or self-administered), whereas assessment is typically a more comprehensive clinical examination such as neuropsychological testing. As there are many cognitive tests included and discussed in the findings, a clear indication that they fall under screening as opposed to a more comprehensive assessment would aid with clarity. A brief definition of how the concept of screening or assessment is interpreted in the introduction or methods section as well as consistent use of these terms throughout the manuscript would aid clarity.

RESPONSE: Excellent, thank you. I have now distinguished between the terms in the introduction, in describing the value of an efficient screen, and have reviewed the manuscript to consistently employ 'screen' in reference to the goal of non-clinical screening to prompt for formal assessment.

 Specific feedback by section:

COMMENT: Line 105: The authors note existing reviews that have examined brain injury/cognitive impairment history in homeless persons. Two of these reviews (Stubbs 2020 and Stone 2019) are quite recent, and both of them report on tools used for screening in homeless individuals. I would suggest briefly discussing the findings of these reviews as pertains to the types of tools/strategies reported on as this would lay more of a foundation for your research questions as well as give you an opportunity to tease out in a bit more detail how your review is addressing a gap that has not been dealt with in these recent reviews (e.g. your focus on tools that can be easily administered by non-clinical staff in a shelter/housing service environment, you are identifying the evidence on acceptability of these screeners for clinicians/staff as well as service users). This might help to highlight the unique contribution of this scoping review as well as its relevance to the intended scientific/clinical audience with more clarity.

RESPONSE: This has now been addressed, and specifically discussed in the final paragraph of the introduction

COMMENT: Lines 108-116: This section of the introduction is a bit challenging to read, potentially due to the in-text list followed immediately by the bullet-point list. It is not quite clear if the bullet point list is among the objectives for the scoping review or if it is better placed in a separate sentence highlighting what specifically will be examined as relates to the viable tools for screening.

RESPONSE: Thanks! Separated the previous works from the objective, and separated into a separate sentence with points pertaining to viability, as suggested.

COMMENT: Table 1 and Section 2.3: It would be helpful to include a rationale for the choice of age limit (i.e. why youth under 16 who are homeless are not included in the potential population).

RESPONSE: Thank you. There are a two reasons for focusing on 16+. One is the use of different assessment tools, and another is the capacity to sign a tenancy or licence agreement. We have provided the rationale in section 2.3. and included recommendations for future research in this area to the conclusion.

COMMENT: Section 3.2.1: It would be helpful to summarize the different types of settings beyond services that are or aren’t exclusive to homeless persons. For example, how many studies were in settings that were general shelters, shelter-embedded substance use or mental health programs, inner city health programs, drop-in clinics for individuals who are marginally housed/homeless, etc. The setting type provides some context for where screening can feasibly happen.

RESPONSE: Thanks! These details have been included: “Ninety-four studies were conducted in settings exclusive to people who are homeless: crisis accommodation or shelter environments (n = 42), generalist homelessness or outreach support settings (N = 31), post-crisis housing support (N = 11), and health settings for people who are homeless (N = 10). Fourteen studies were conducted in other health or community settings that service both people who are and who are not homeless, including general health or hospital settings (n = 7), mental health or substance use treatment programs (n = 5), and diverse community or social welfare settings (n = 2).”

COMMENT: Table 2: For clarity, a definition or age range for ‘older people’ and ‘younger people’ would be helpful.

RESPONSE:

Age ranges now included in table 2:

Older people (50+)

Young people (16-24)

COMMENT: Lines 300-303 and Table 3. What was the further investigation done to determine the administration requirements and costs for each measure? For example, did you review the developers’ websites or publications on each instrument? For clarity in table 3, it would be important to demonstrate where the reported information is coming from, if not directly from the publications included in the review. If possible, based on your investigations, some greater clarity regarding what is meant by ‘specialist’ here would be helpful (i.e. any registered clinician vs a neuropsychologist, vs any worker with specific training, etc).

RESPONSE: Addition: “Further investigation was undertaken to determine whether instruments required specialist qualifications to administer, through first searching Pearson Assessments [67] or PAR Inc. [68] for qualification grade, and if unavailable on these platforms, through developer websites. This process determined that 14 of the 17 cognitive screening tools require specialist qualifications to administer (psychologist, occupational therapist, or "health professional"), whilst the remaining 3, as well as the 3 TBI screens, are not restricted.”

COMMENT: Discussion section: Some additional elaboration on the future research directions as relates to the review findings would strengthen the discussion. The conclusion recommends identifying common factors among persons experiencing homelessness and ABI/cognitive impairment as well as consultation regarding facilitators and barriers to screening – incorporating some elaboration of/rationale for these recommendations with reference to the literature and review findings within the earlier discussion section would be helpful.

RESPONSE: This is actually just a practice recommendation, as existing data that is collected may, in combination, also flag possible impairment and benefit from referral. Amended to: As application of brief screening instruments alone may not capture all those benefiting from more comprehensive assessment and support, identification of other common factors amongst this cohort may be helpful to enhance screening protocols and provide further indicators of potential impairment and referral needs in this group.”

REVIEWER 3

COMMENT:

 To identify the need to understand the intersection between cognitive impairment/TBI and the homeless population

 To identify possible screening/Ax tools for homeless services to adopt as routine practice.

 2.1 Identification of the Research Questions Lines 128- 137 relevant and answered. However note that the term “brain injury” is used in 2 of the 3 questions, not TBI. See highlighted yellow comment below.

RESPONSE: Thank you, now resolved as per above responses to comments from other reviewers.

o Overview the manuscript is clear and comprehensive and relevant to the field of homeless/cognitive impairment/TBI.

o Detailed analysis of the 108 publications with the finding 151 instruments of cognitive function and 8 for TBI.

COMMENT:

ï‚· General concept comments Article: highlighting areas of weakness, the testability of the hypothesis, methodological inaccuracies, missing controls, etc.

o The abstract mentions keywords, including social determinations of health, however this is not linked throughout the manuscript. I would recommend strengthening this where outcomes are mentioned in the introduction between Lines 42 – 83.

RESPONSE: Thank you - explicit description of social determinants affected included in early paragraph introducing the issue (formerly beginning line 42) and reference to review completed by Stone et al and link to consistent link to impact on socioeconomic factors.

COMMENT: o There is a difference between TBI and ABI even though these terms are often used interchangeably. I recommend given that Appendix A Table A1 Search strategy p 16, as shown below, has ABI as the heading that the difference is identified and that the paper is focused on both TBI and ABI.

RESPONSE: This has been resolved, changing language throughout, and explicitly describing the issue of TBI-specific instruments in the literature generated in this review.

ï‚·

ï‚· Specific comments Repetitiveness/duplication highlighted below + specific comments:

COMMENT: High rates of cognitive impairment [5, 9-11], TBI [12-27] or both [7, 28, 29] have 42

been found in homeless populations. Failure to identify cognitive impairment and TBI 43

amongst people experiencing homelessness leads to poor housing outcomes for individ- 44

uals in this cohort, places undue pressure on the homelessness service delivery system 45

and increases demand on emergency and other public service responses. 46

RESPONSE: Thank you! Repetition of 'cognitive impairment and TBI' replaced with 'these issues'

COMMENT: Reference to 'individual' does this mean and individual without cognitive impairment and/or TBI?

In addition to poor outcomes for the individual, inadequate engagement with 74

planned health and welfare services, and inefficient utilisation of the homelessness ser- 75

vice system, long-term homelessness is associated with increased use of emergency and 76

other public resources. People experiencing homelessness, cognitive impairment and 77

TBI are more likely to have contact with police, fire service and paramedics, increased 78

presentations to emergency departments, greater frequency of hospital admissions, high 79

use of emergency housing and welfare services, and are more likely to be in contact with 80 3

state justice and child protection [24, 46, 47]. A similar profile exists for people with cog- 81

nitive impairment who are housed with inadequate support [48]. As such, there is a 82

strong economic argument for disrupting the cycle of homelessness through prompt 83 identification of cognitive impairment and TBI.

RESPONSE: The referenced literature in the paragraph talking about this relates to general, long-term homelessness (which in itself has a high population of people with cognitive impairment/TBI). The point here is to pull together the earlier description of health impacts at the individual level, alongside disengagement at the planned service level, and how this naturally results in increased use of emergency services. To reduce confusion, I have replaced 'poor outcomes for the individual' with 'poor individual outcomes'.

COMMENT: Recommend “This would support long-term and repeat users of homelessness services to access service environments that have capability and capacity to understand the impacts of cognitive impairment/TBI on functional impairments.”

This would redirect long-term and repeat users of homelessness services into 87

more adequate service environments, add new capacity to homelessness programs, and 88

reduce unplanned use of other public resources

RESPONSE: Replaced. Thank you.

COMMENTS:

ï‚· Is the manuscript clear, relevant for the field and presented in a well-structured manner?

o Yes

ï‚· Are the cited references mostly recent publications (within the last 5 years) and relevant? Does it include an excessive number of self-citations?

o Yes, noting that cited references also include publications from 2001 onwards. However, this is relevant given the scarcity of academic publications in this area.

ï‚· Is the manuscript scientifically sound and is the experimental design appropriate to test the hypothesis?

o Yes

ï‚· Are the manuscript’s results reproducible based on the details given in the methods section?

o Yes

ï‚· Are the figures/tables/images/schemes appropriate? Do they properly show the data? Are they easy to interpret and understand? Is the data interpreted appropriately and consistently throughout the manuscript? Please include details regarding the statistical analysis or data acquired from specific databases.

o Overall, the presentation, the data, and the interpretation and understanding are outstanding.

o The conclusion is consistent with the evidence and arguments presented and is supported by the listed citations and are logical recommendations.

RESPONSE: Thank you for these very encouraging responses.

Use of the term brain injury not Traumatic brain injury.

Further investigation to identify potential other common factors amongst people ex- 445

periencing homelessness who have a known cognitive impairment or brain injury may 446

further enhance screening for potential impairment in this group

RESPONSE: We have attempted to address this by including the more general term “brain injury” (which was included as a search term and we attempted to include this in the included screening instruments) and highlighting that only TBI screening tools were included in the final set of instruments.

COMMENTS:

Rating the Manuscript

During the manuscript evaluation, please rate the following aspects:

ï‚· Novelty:

o The question is original and well defined and provides an advancement on current knowledge

ï‚· Scope: Does the work fit the journal scope*?

o Yes

ï‚· Significance:

o The results are appropriately interpreted. The recommendations are appropriate for further examination.

ï‚· Quality

o The presentation of the results is outstanding, and the article is well written with minor edits to occur.:

ï‚· Scientific Soundness:

o The scoping review is the most appropriate study design to answer the questions. Database service, search terms, hand search, data mapping and use of the PRISMA used to the highest standards

ï‚· Interest to the Readers:

o The conclusions will attract a wide readership not just researchers but also for policy makers given that there is a significant concern both in Australia and globally around housing and homeless and disability.

  • Overall Merit:

o There is merit in publishing this manuscript as detailed in the

interest to the readers.

ï‚· English Level: Is the English language appropriate and understandable?

o Yes

RESPONSE: Thank you once again for these very favourable and encouraging comments.

Reviewer 2 Report

This very interesting review focused on identifying tools and strategies used to identify brain injury and cognitive impairment. There was an emphasis on understanding what tools exist that can be administered by non-clinician/specialist service providers in homeless service settings to expedite referrals to specialty services. This review identifies some important gaps in our current knowledge regarding screening for brain injury or cognitive impairments in homelessness service settings, particularly regarding the validity and acceptability of screening tools in these settings and provides a foundation upon which further research can be built. A scoping review was an appropriate choice to examine the identified research objectives, and the methodology was guided by an established framework (Arksey and O’Malley) for conducting scoping reviews and this report also adhered to PRISMA reporting guidelines. The review adheres closely to these guidelines for robust reviews.

Feedback for manuscript overall. My main feedback/concerns overall are related to clarification of terms and concepts used throughout the report. Specifically:

Cognitive impairment has a wide scope as a concept and it is not clear in the introduction as a reader why CI is being examined along with TBI specifically, as opposed to cognitive impairment along with acquired brain injury more broadly. There is also reference to acquired brain injury or brain injury more generally in other sections of the manuscript, including the inclusion criteria. As homeless persons are at high risk for non-traumatic ABI in addition to TBI (e.g. hypoxic brain injury related to substance overdose) some clarity about the focus on TBI or whether all acquired brain injuries fell within the scope of this review is needed.

The terms screening, assessment, and measurement are used throughout the manuscript and I wonder if some clarity about the concept of screening versus assessment might be helpful. Screening would appear to be the focus of this review (ie. a brief test for early identification of an issue, often part of routine care, can be administered by non-clinician staff with appropriate training or self-administered), whereas assessment is typically a more comprehensive clinical examination such as neuropsychological testing. As there are many cognitive tests included and discussed in the findings, a clear indication that they fall under screening as opposed to a more comprehensive assessment would aid with clarity. A brief definition of how the concept of screening or assessment is interpreted in the introduction or methods section as well as consistent use of these terms throughout the manuscript would aid clarity.

Specific feedback by section:

Line 105: The authors note existing reviews that have examined brain injury/cognitive impairment history in homeless persons. Two of these reviews (Stubbs 2020 and Stone 2019) are quite recent, and both of them report on tools used for screening in homeless individuals. I would suggest briefly discussing the findings of these reviews as pertains to the types of tools/strategies reported on as this would lay more of a foundation for your research questions as well as give you an opportunity to tease out in a bit more detail how your review is addressing a gap that has not been dealt with in these recent reviews (e.g. your focus on tools that can be easily administered by non-clinical staff in a shelter/housing service environment, you are identifying the evidence on acceptability of these screeners for clinicians/staff as well as service users). This might help to highlight the unique contribution of this scoping review as well as its relevance to the intended scientific/clinical audience with more clarity.

Lines 108-116: This section of the introduction is a bit challenging to read, potentially due to the in-text list followed immediately by the bullet-point list. It is not quite clear if the bullet point list is among the objectives for the scoping review or if it is better placed in a separate sentence highlighting what specifically will be examined as relates to the viable tools for screening.

Table 1 and Section 2.3: It would be helpful to include a rationale for the choice of age limit (i.e. why youth under 16 who are homeless are not included in the potential population).

Section 3.2.1: It would be helpful to summarize the different types of settings beyond services that are or aren’t exclusive to homeless persons. For example, how many studies were in settings that were general shelters, shelter-embedded substance use or mental health programs, inner city health programs, drop-in clinics for individuals who are marginally housed/homeless, etc. The setting type provides some context for where screening can feasibly happen.

Table 2: For clarity, a definition or age range for ‘older people’ and ‘younger people’ would be helpful.

Lines 300-303 and Table 3. What was the further investigation done to determine the administration requirements and costs for each measure? For example, did you review the developers’ websites or publications on each instrument? For clarity in table 3, it would be important to demonstrate where the reported information is coming from, if not directly from the publications included in the review. If possible, based on your investigations, some greater clarity regarding what is meant by ‘specialist’ here would be helpful (i.e. any registered clinician vs a neuropsychologist, vs any worker with specific training, etc).

Discussion section: Some additional elaboration on the future research directions as relates to the review findings would strengthen the discussion. The conclusion recommends identifying common factors among persons experiencing homelessness and ABI/cognitive impairment as well as consultation regarding facilitators and barriers to screening – incorporating some elaboration of/rationale for these recommendations with reference to the literature and review findings within the earlier discussion section would be helpful.

Author Response

(The authors gave the same response as above.)

Author Response

(The authors gave the same response as above.)
